

# Two predicted models based on ceRNAs and immune cells in lung adenocarcinoma

Miaomiao Zhang[1,\*], Peiyan Zheng[2,\*], Yuan Wang[1] and Baoqing Sun[2]

[1] The First Affiliated Hospital of Guangzhou Medical University, Guangzhou Institute of Respiratory Diseases, State Key Laboratory of Respiratory Disease, Guangzhou, China

[2] Department of Allergy and Clinical Immunology, Guangzhou Institute of Respiratory health, State Key Laboratory of Respiratory Disease, National Clinical Research Center of Respiratory Disease, First Affiliated Hospital of Guangzhou Medical University, State Key Laboratory of Respiratory Disease, Guangzhou, China

[\*] These authors contributed equally to this work.

## ABSTRACT

**Background**. It is well accepted that both competitive endogenous RNAs (ceRNAs) and immune microenvironment exert crucial roles in the tumor prognosis. The present study aimed to find prognostic ceRNAs and immune cells in lung adenocarcinoma (LUAD).

**Materials and Methods**. More specifically, we explored the associations of crucial ceRNAs with the immune microenvironment. The Cancer Genome Atlas (TCGA) database was employed to obtain expression profiles of ceRNAs and clinical data. CIBERSORT was utilized to quantify the proportion of 22 immune cells in LUAD.

**Results**. We constructed two cox regression models based on crucial ceRNAs and immune cells to predict prognosis in LUAD. Subsequently, seven ceRNAs and seven immune cells were involved in prognostic models. We validated both predicted models via an independent cohort GSE72094. Interestingly, both predicted models proved that the longer patients were smoking, the higher risk scores would be obtained. We further investigated the relationships between seven genes and immune/stromal scores via the ESTIMATE algorithm. The results indicated that CDC14A and H1F0 expression were significantly related to stromal scores/immune scores in LUAD. Moreover, based on the result of the ceRNA model, single-sample gene set enrichment analysis (ssGSEA) suggested that differences in immune status were evident between high- and low-risk groups.

Corresponding authors
Yuan Wang,
YuanWangxz1992@126.com
Baoqing Sun,
sunbaoqing@vip.163.com

## INTRODUCTION

Lung cancer is considered as the most common cause of survival-associated cancer globally (*Bray et al., 2018*). Besides, lung adenocarcinoma (LUAD) approximately accounts for 50% of all lung cancer types (*Imielinski et al., 2012*). Although the improvement of diagnosis and the comprehensive treatment in the lung cancer has been made, the 5-year overall survival rate for lung cancer is still approximately 18%, mainly due to invisible early symptoms

(*Siegel, Miller & Jemal, 2018*). Thus, it is of great necessity to identify sensitive and accurate biomarkers in the early stage.

Competitive endogenous RNAs (ceRNAs), consisting of a group of regulatory RNA molecules, bind with the specific miRNAs and then affect the protein level (*Kartha & Subramanian, 2014*; *Phelps et al., 2016*; *Shukla, Singh & Barik, 2011*). Previous studies have shown that ceRNA regulatory networks play essential roles in tumor progression and migration (*Bartel, 2009*; *Qi et al., 2015*; *Sanchez-Mejias & Tay, 2015*; *Tay, Rinn & Pandolfi, 2014*). Besides, increasing studies have demonstrated that immune microenvironment (TIME) was also associated with the prognosis in various kinds of tumors, including LUAD (*Lavin et al., 2017*; *Liu, Niu & Qiu, 2020*). Stromal cells and immune cells constitute the central part of TIME. Stromal cells participate in the process of tumorigenesis and metastasis and are generally conducive to the movement of cancer cells (*Ni et al., 2018*). Additionally, various studies revealed that immune cells exhibit large effects on clinical outcomes in many cancers. Donnem et al. summarized that different types of immune cells affected the clinical stage and prognosis in non-small lung cancer (*Donnem et al., 2016*; *Liu et al., 2020*). As a result, to further investigate potential biomarkers based on immune cells and ceRNAs in LUAD, we constructed Cox proportional hazards models and predicted nomograms.

However, the mechanisms of ceRNAs in the tumor microenvironment still remain unclear in LUAD. It is well accepted that cancer stem cells (CSCs) are tumorigenic cells with self-renewal ability. Besides, CSCs promote tumor immune to escape via expressing a variety of immune factors (*Bruttel & Wischhusen, 2014*). In the current study, RNA stem scores and DNA stem scores were applied to assess tumor stem cell content (*Malta et al., 2018*). ESTIMATE algorithm was utilized to evaluate the stromal and immune cells in every sample (*Yoshihara et al., 2013*). Thus, we explored the correlation between crucial prognostic genes with RNAss/DNAss scores and stromal/immune scores. All in all, in this study, we comprehensively analyze the associations between crucial ceRNAs and immune infiltration landscape, crucial ceRNAs and tumor stem cells, which may offer an important perspective for early diagnosis and treatment of LUAD.

## MATERIALS AND METHODS

### Identification of differentially expressed ceRNAs

In the present study, expression profiles of LUAD and the corresponding clinical information were derived from The Cancer Genome Atlas (TCGA) database, including lncRNA, miRNA, and mRNA. The R package "DEseq2" was used to screen the differentially expressed mRNAs, miRNAs and lncRNAs (DEmRNA, DEmiRNA and DElncRNA) between normal and tumor samples (*Love, Huber & Anders, 2014*). Specifically, a false discovery rate (FDR) $<0.05$ and log2 |fold change |$> 1$ were set as cutoff criteria.

### ceRNA network construction

Both lncRNA-miRNA and miRNA-mRNA interactions were predicted using, miRcode and starBase (*Chou et al., 2018*; *Jeggari, Marks & Larsson, 2012*; *Li et al., 2014*). DEmiRNAs, DEmRNAs, and DE lncRNAs, which presented a significant difference in hypergeometric

distribution detection and correlation analysis ($P < 0.05$), were selected to construct a lncRNA-miRNA-mRNA regulatory network and then was visualized by Cytoscape 3.80 (*Shannon et al., 2003*).

## Construction of nomogram based on key ceRNAs

Univariate Cox analysis was employed to screen the survival-associated ceRNAs. Lasso regression analysis was applied to ensure that the Cox model was not overfitted. Eventually, the candidate ceRNAs were integrated to establish the multivariate cox regression models. Risk scores were described as following: risk score=(Exp RNA1* $\beta$1) + (Exp RNA2* $\beta$2 ) + ... + (Exp RNAn* $\beta$n). It is worth noting that Exp stands for the expression level, and $\beta$ denotes the regression coefficient. Based on the risk scores, patients with LUAD were divided into high-risk and low-risk groups. The corresponding Kaplan–Meier survival curves were carried out to reveal the overall survival (OS) within different groups. In addition, we also performed the ROC curve to evaluate the specificity and sensitivity of the model. Finally, according to the multivariate model result, we constructed a nomogram to predict patients' prognostic values. The calibration curve was applied to evaluate the accuracy of the nomogram.

## Construction of nomogram based on immune cells

With a good deconvolution performance in gene expression profiles, CIBERSORT method could estimate the proportion of particular cell types (*Newman et al., 2015*). In this study, CIBERSORT algorithm was utilized to estimate 22 tumour-infiltrating immune cells in LUAD by collating and calculating the genes expression in the tumor samples in the TCGA database. Samples with a CIBERSORT output of $p < 0.05$ presented that the predicted proportion of immune cells were correct.

The Wilcoxon test was conducted to examine the significant difference of immune cells between the tumor samples and healthy samples. We also performed multivariate Cox regression and produced the corresponding risk scores. Based on the multivariate Cox model results, we finally built a predicted nomogram. Similar to ceRNAs, patients were also stratified into high-risk and low-risk groups on the basis of the mean risk score. Subsequently, the prognostic model's sensitivity and specificity were examined by the ROC curve, and the calibration curve investigated the accuracy of the nomogram.

## Validation of prognostic equations with GEO

To better elucidate the accuracy of both multivariate Cox models, an external validation dataset GSE72094 ($n = 442$) was applied in the present study. We selected this dataset because it contained most LUAD patients with clinical data in the Gene Expression Omnibus (GEO) database. Based on the median risk score generated from TCGA, we also classified patients with high- and low-risk patients. K-M and ROC curves were applied to evaluate the prognostic efficacy of multivariate models in TCGA.

## Estimation of tumor stemness and microenvironment

Stemness scores of "RNAss" and "DNAss" referred to the result based on mRNA and DNA-methylation respectively. The larger stemness scores were, the higher likelihood

more stem cells were infiltrating. ESTIMATE algorithm could speculate the level of stromal cell, immune cells and tumor purity, respectively (*Yoshihara et al., 2013*). Correspondingly, stromal/immune/Estimate scores were significantly associated with stromal cells, immune cells and tumor purity, respectively. We utilized the "ESTIMATE" package to assess immune, stromal scores, and the sum of both in individual patients. The higher scores, the higher proportion of corresponding cells were. Additionally, the correlation of hub ceRNAs with above scores was examined by Spearman analysis. The ssGSEA method could apply expressed traits of immune cell population to individual tumor samples (*Barbie et al., 2009*). Based upon the results of ssGSEA, we calculated infiltrating level of immune cells and immune-associated functions in LUAD samples by using the "gsva" package (*Hänzelmann, Castelo & Guinney, 2013*). Furthermore, we compared the differences in these immune data sets between the high-risk group and the low-risk group.

## Statistical analysis

All statistical analyses were accomplished with R software 4.0.2. The Wilcoxon rank sum tests were conducted for comparisons between two groups. Only a two-sided $P$-value <0.05 was recognized as statistically significant.

# RESULTS

Clinicopathological characteristics of LUADClinicopathological characteristics in the TCGA database and GSE72094 are displayed in Tables 1 and 2, respectively. In addition, complete clinical information for each patient is listed in Table S1.

## Identification of DEmRNAs, DEmiRNAs and DElncRNAs

Figure 1 presents our workflow for the bioinformatic analysis. Among the whole expression profiles, totally 1645 upregulated DEmRNAs, and 1,344 downregulated DEmRNA were screened out (Figs. 2A and 2B). Furthermore, we also identified 163 upregulated DElncRNA, 52 downregulated DElncRNA (Figs. 2C and 2D), 111 upregulated DEmiRNA, and 87 downregulated DEmiRNA (Fig. 3B). Specifically, the top 20 significantly upregulated- and downregulated- miRNAs were shown in Fig. 3A.

## Construct Nomogram of ceRNAs

To investigate the endogenously competitive relationships between the lncRNA and mRNA, we constructed an intersected ceRNA network. Seven lncRNAs (H19, AC074117.1, FBXL19-AS1, MAGI2-AS3, SNHG3, PVT1 and SNHG1), 94mRNA, and 15miRNA were involved in the network (Table S2). Hsa-miR-130b-3p and hsa-miR-29b-3p were the top 2 regulated miRNA in the ceRNA network (Fig. 4). In addition, the initial univariate Cox regression (Fig. S1) and Lasso regression analysis were further conducted to investigate the key biomarkers (Figs. 5B and 5C). Seven survival-associated biomarkers were finally involved in a new multiple Cox regression analysis (Fig. 5A). According to the result, all biomarkers belonged to protein-coding RNAs. Consequently, the corresponding risk score was generated (Table 3): risk score = [Exp CDC14A * (-0.281888)]+(Exp LOXL2*0.114969) + (Exp CCT6A * 0.174244)+(Exp E2F7 *0.187164)+(Exp GPR37*0.099737)+(Exp H1F0 *0.197590)+(Exp SMOC1 * 0.068385).

**Table 1  Clinicopathological characteristics baseline in LUAD in TCGA.**

| Characteristics | | Number | (%) |
|---|---|---|---|
| Total | | 522 | 100% |
| Age | | | |
| | >65 | 262 | 50.2% |
| | <=65 | 241 | 46.2% |
| | Unknown | 19 | 3.6% |
| Gender | | | |
| | Female | 280 | 53.6% |
| | Male | 242 | 46.4% |
| Vital status | | | |
| | Dead | 188 | 36.0% |
| | Alive | 334 | 64.0% |
| Clinical stage | | | |
| | I | 279 | 53.4% |
| | II | 114 | 21.8% |
| | III | 85 | 16.3% |
| | IV | 26 | 5.0% |
| | Unknown | 8 | 1.5% |
| T stage | | | |
| | T1 | 172 | 33.0% |
| | T2 | 281 | 53.8% |
| | T3 | 47 | 9.0% |
| | T4 | 19 | 3.6% |
| | Unknown | 3 | 0.6% |
| M stage | | | |
| | M0 | 353 | 67.6% |
| | M1 | 25 | 4.8% |
| | Unknown | 144 | 27.6% |
| N stage | | | |
| | N0 | 335 | 64.2% |
| | N1 | 98 | 18.8% |
| | N2 | 75 | 14.4% |
| | N3 | 2 | 0.4% |
| | Unknown | 12 | 0.2% |
| Smoking status | | | |
| | <=20 | 45 | 8.6% |
| | >20 | 139 | 26.6% |
| | Unknown | 338 | 64.80% |

**Notes.**
LUAD, lung adenocarcinoma.

As shown in Fig. 5D, the prognostic score could be a promising indicator to distinguish the LUAD patients according to the individual prognosis ($P = 5.701e{-}8$). The area under ROC reflected the predictive efficacy of the prognostic signature (AUC of 3-year

**Table 2  Clinicopathological characteristics baseline in LUAD in GSE72094.**

| Characteristics | | Number | (%) |
|---|---|---|---|
| Total | | 442 | 100% |
| Age | | | |
| | >65 | 294 | 50.2% |
| | <=65 | 127 | 46.2% |
| | Unknown | 21 | 3.6% |
| Gender | | | |
| | Female | 280 | 53.6% |
| | Male | 242 | 46.4% |
| Smoking status | | | |
| | Ever | 335 | 75.8% |
| | Never | 33 | 7.5% |
| | Unknown | 74 | 16.7% |
| Stage | | | |
| | I | 265 | 60.0% |
| | II | 69 | 15.6% |
| | III | 63 | 14.3% |
| | IV | 17 | 3.9% |
| | Unknown | 28 | 6.3% |
| Vital status | | | |
| | Alive | 298 | 67.4% |
| | Dead | 122 | 27.6% |
| | Unknown | 22 | 5.0% |

survival:0.772) (Fig. 5E). By the GEO validation set, the AUC of gene signature at 3-year was 0.711 (Fig. S2A). Additionally, K-M analysis further confirmed the difference between high- and low-risk groups ($P = 9.963e-05$) (Fig. S2B). Eventually, we assessed the prognosis of patients with LUAD by a nomogram (Fig. 6A), and the accuracy of the nomogram was confirmed by the calibration curve (Fig. 6C). As shown in Fig. 7A, based on this prognostic model, the risk score in the group with fewer smoking years ($\leq 20$) was lower than the group with more smoking years ($>20$) ($P = 0.025$).

## Construct nomogram of immune cells

By applying the CIBERSORT, we estimated the percentages of 22 immune cell types in LUAD (Figs. S3B and S3C) (*Chen et al., 2018*). The violin plot indicated that 13 immune cell types showed a significant difference in percentages of immune cells between tumor samples and healthy samples by the Wilcoxon test (Fig. S3D). The correlation analysis of immune cells could be found in Fig. S3A. Subsequently, after the analysis of univariate cox regression and lasso regression, seven immune cells were taken as prognostic biomarkers and eventually integrated into a new multivariate model (Figs. 8A, 8B and 8C). The formula was presented as following (Table 4): risk score = [Exp CD8 *(-3.425649 )]+(Exp Tregs*9.175062 )+[Exp Monocytes*(-5.318492)]+(Exp M1 *6.134551)+(Exp Dendritic cells activated*5.539130 )+(Exp Mast cells activated
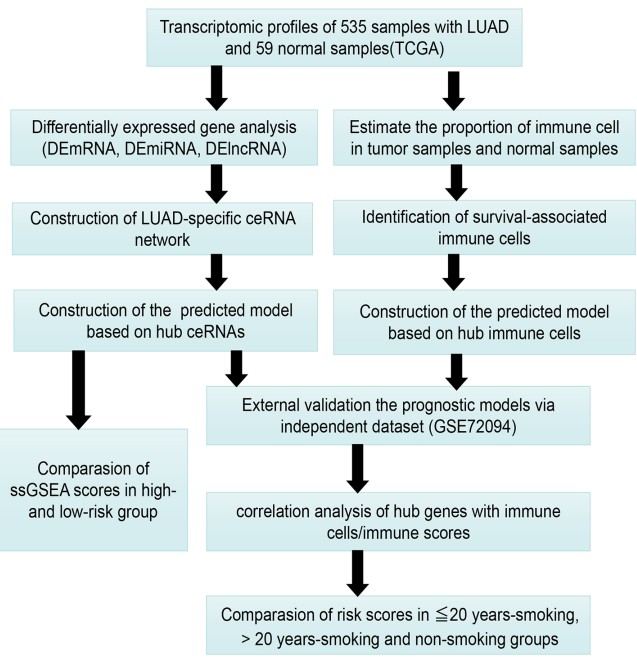

**Figure 1** **The workflow for bioinformatic analysis.**

*24.611110 )+(Exp Eosinophils*37.351722). The K-M analysis results showed a significant difference in overall survival between high- and low- risk groups ($P = 3.342e{-}05$) (Fig. 8D). The ROC curve (AUC of 3-years: 0.766) demonstrated that the multivariate model was recognized as a fair potential to monitor the prognostic efficacy (Fig. 8E). In the external validation set, the model also carried out adequate predicted capacity. The AUC of immune cell signature was 0.738 at 3-years (Fig. S2C). K-M curves reflected that the low-risk group had longer overall survival than the high-risk group ($P = 2.922e{-}04$) (Fig. S2D). A predicted nomogram was also performed, and the discrimination was conducted to test the calibration quality (Figs. 6B and 6D). Similarly, we could also found that the longer smoking history was related to the higher the risk scores based on the predicted model of immune cells in smoking groups ($P = 0.0046$) (Fig. 7B). The correlations of key prognostic biomarkers in the two models illustrate that the expression of E2F7 is positively related to Macrohages M1 ($R = 0.4$) (Fig. S4).

Besides, the Univariate and multivariate survival analyses revealed that risk scores in the two models were both independent prognostic factors (Fig. S5).

## Correlation of hub ceRNAs expression with tumor stemness and microenvironment

As shown in Fig. 9, CCT6A, H1F0 and E2F7 were positively related to RNAss and DNAss ($R = 0.22$ to $0.53$). A significant negative association was detected between CDC14A and RNAss ($R = -0.25$, $P = 7.5e{-}8$). Nevertheless, we did not find a significant correlation between LOXL2, GPR37 and SMOC1 with tumor stemness. In addition, H1F0 displayed a negative association with stromal scores, immune scores and estimate scores ($R = -0.34$

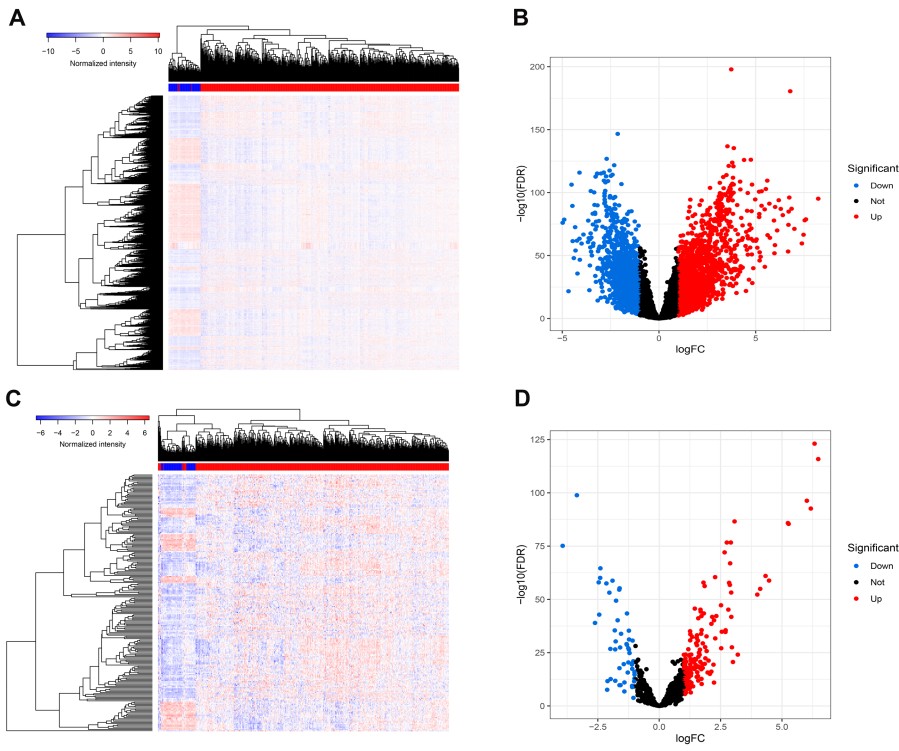

**Figure 2  Differentially expressed mRNAs and lncRNA.** Heatmap (A) and volcano plot (B) of DEm-RNA.Heatmap (C) and volcano plot (D) of DElncRNA. Red dots and green dots represent up-regulated, down-regulated differentially expressed genes, respectively. DEmRNA and DElncRNA represent differentially expressed mRNA and lncRNA, respectively.

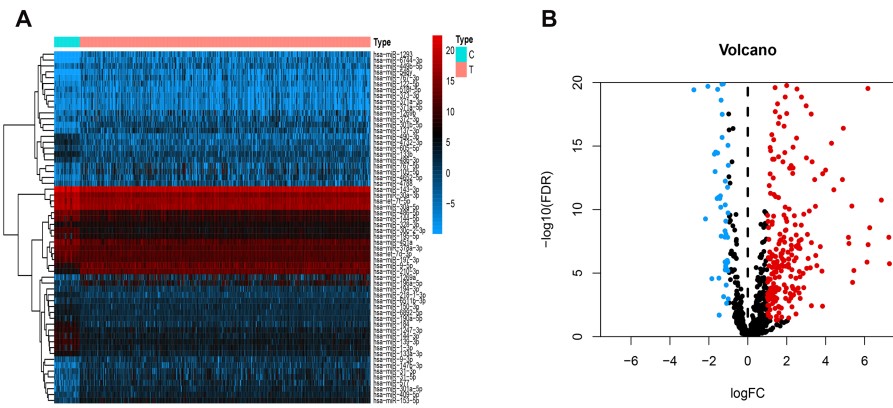

**Figure 3  Differentially expressed miRNAs.** Heatmaps of the top 20 differentially upregulated and the top 20 differentially downregulated miRNAs (A). Volcano plot (B) of DEmiRNAs. Red dots and green dots represent up-regulated, down-regulated differentially expressed genes, respectively. DEmiRNAs, differentially expressed miRNAs.

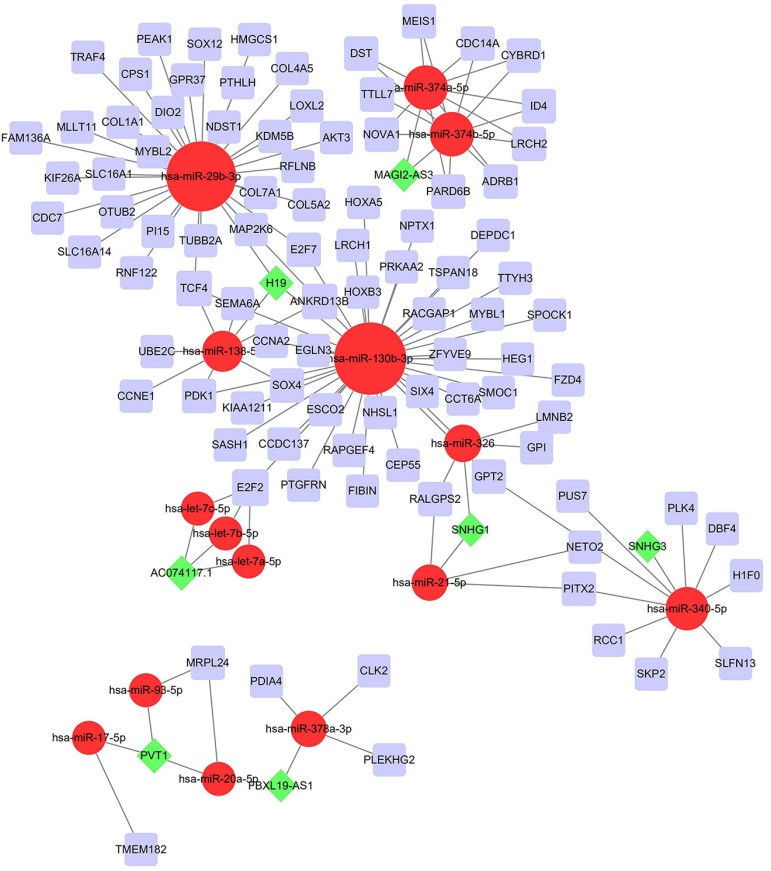

**Figure 4 Construction of the ceRNA network in LUAD.** Red circles, purple squares, green diamonds represent miRNA, mRNA and lncRNA, respectively. Node sides of miRNA indicate the number of junction nodes. The greater the node sizes, the more junction nodes.

to −0.27). However, CDC14A showed a positive association with the microenvironment-related scores ($R = 0.31$ to $0.35$).

## Multiple databases validation

To validate CDC14A and H1F0 expression in LUAD, we utilized multiple databases. Based on the result of UCLAN, GEPIA and TIMER, it could be concluded that H1F0 was highly expressed while CDC14A was weakly expressed in LUAD (Fig. S6). Furthermore, the CDC14A and H1F0 protein expression were obtained from The Human Protein Atlas data (HPA), indicating that H1F0 was high staining in tumor cells, while it was medium staining in normal tissues (Figs. S7A and S7B). The opposite trend was observed in CDC14A. Besides, the association of H1F0 and CDC14A with immune cells were validated in TISIDB. CDC14A expression was positively associated with Immature B cell ($R = 0.345$), NK cell ($R = 0.308$), Eosinophils ($R = 0.357$), and Mast cell ($R = 0.332$) (Fig. S8A–S8E). Controversially, H1F0 expression was negatively related to Eosinophil ($R = −0.34$), Macrophage ($R = −0.359$), NKT ($R = −0.323$), Tem_CD8 ($R = −0.323$) and Th1 ($R = −0.407$) (Figs. S8F–S8J).

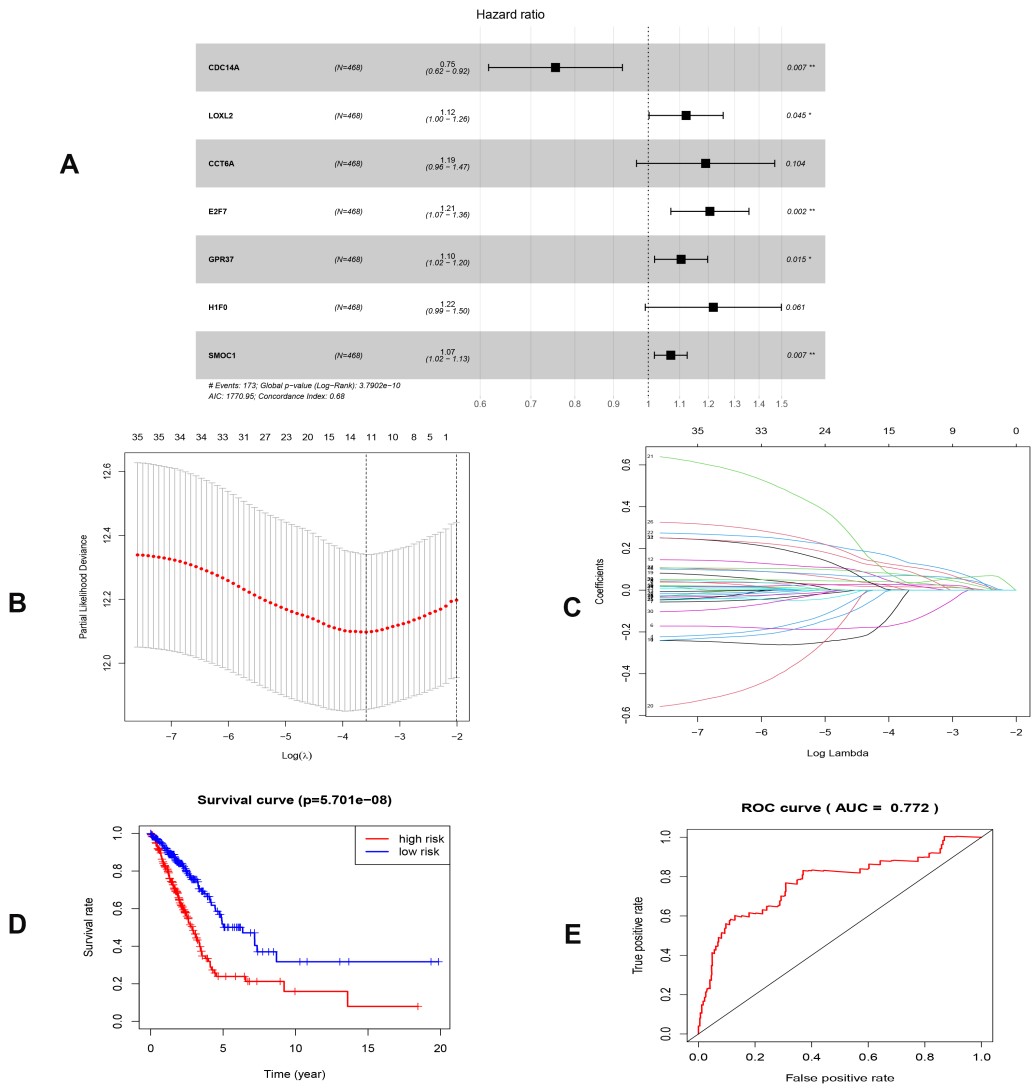

**Figure 5** **Construction of the prognostic model involving the key RNAs of the ceRNA network in LUAD.** (A), multivariate Cox regression. (B), (C) lasso regression. (D) the Kaplan–Meier survival curve. (E) the ROC curve. ROC, the Receiver Operating Characteristic. LUAD, lung adenocarcinoma.

## Differences in ssGSEA scores in TCGA/GEO cohort

To further elucidate the immune status in risk- and low groups of ceRNA signatures, we performed the ssGSEA algorithm to estimate the immune infiltrating between two groups (Fig. S9). Of note, low-risk group were associated with higher scores of aDCs, B_cells, CD8+_T_cells, iDCs, Mast_cells, Neutrophils, pDCs, T_helper_cells and TIL (Figs. 10A and 10B). Similarly, in the lower-risk group, the scores of Check-point, HLA, T_cell_co-stimulation and Type_II_IFN_Reponse in still higher in the low-risk group.Additionally, the differences above were confirmed by the GEO cohort (Figs 10C and 10D).

**Table 3** **Multivariate Cox proportional hazards regression model based on ceRNAs in LUAD.**

| Genes | Coef | HR | 95% CI | *P* value |
|---|---|---|---|---|
| CDC14A | −0.282 | 0.754 | (0.616–0.924) | 0.007 |
| LOXL2 | 0.115 | 1.122 | (1.002–1.256) | 0.045 |
| CCT6A | 0.174 | 1.190 | (0.965–1.468) | 0.104 |
| E2F7 | 0.187 | 1.206 | (1.071–1.358) | 0.002 |
| GPR37 | 0.100 | 1.105 | (1.019–1.198) | 0.015 |
| H1F0 | 0.198 | 1.218 | (0.991–1.498) | 0.061 |
| SMOC1 | 0.068 | 1.071 | (1.019–1.125) | 0.007 |

**Notes.**

LUAD, lung adenocarcinoma; HR, hazrd ratio; CI, confidence interval.

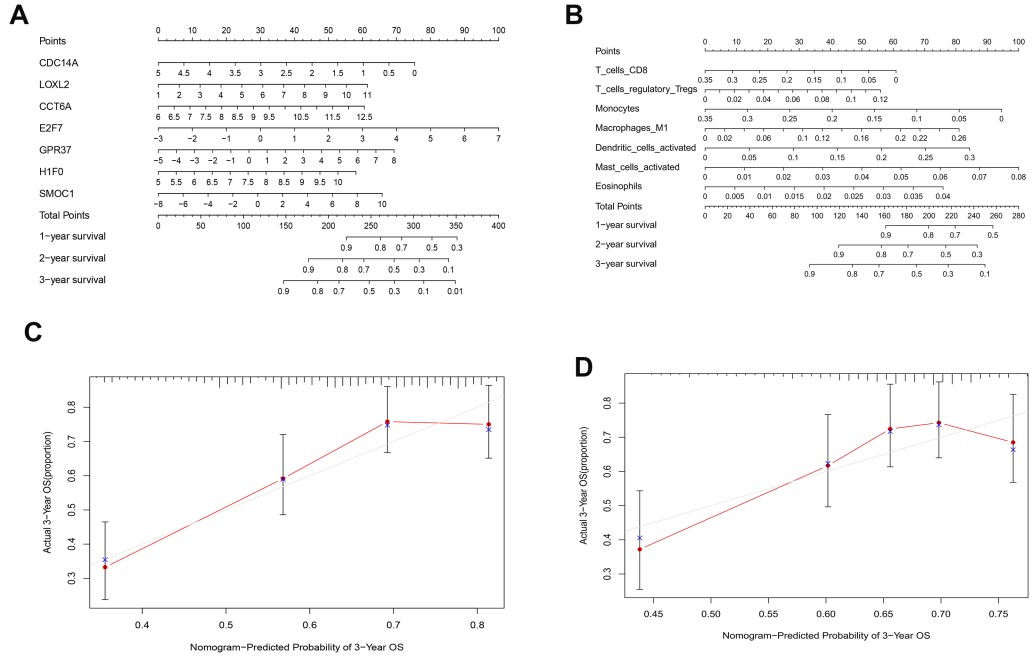

**Figure 6** **Nomograms based on key ceRNAs and immune cells.** Nomograms predicting 1-, 2- and 3-year overall survival of ceRNAs signatures (A) and immune cells signatures (B). Calibration curves of the nomogram at the 3-year of ceRNAs (C) signatures and immune cells signatures (D).

## DISCUSSION

With the reduced overall survival rate, lung adenocarcinoma is the most common pathological type since most patients were diagnosed at an advanced stage. Currently, there is a general consensus that both molecular and cellular characteristics play an imposing role in oncogenesis and metastasis, and thus they were suggested as underlying prognostic signatures. However, the integrative analysis of ceRNAs and microenvironments has yet to be fully explored. Previously, there were several studies that explored potential biomarkers of diagnosis and prognosis in LUAD via constructing the ceRNA model. Sui et al. constructed lncRNA-related network of LUAD with 29 lncRNAs,24 miRNAs

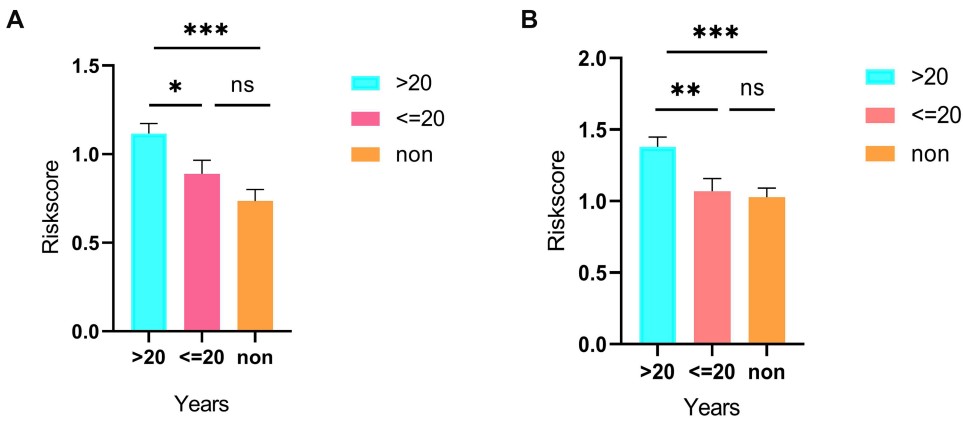

**Figure 7** Differences in risk scores between >20-smoking years, ≤20-smoking years and non-smoking based on ceRNAs (A) and immune cells models (B).

and 72mRNA (*Sui et al., 2016*). Then, they investigated the relevance of lncRNA and clinical characteristics. *Cao et al. (2020)* established a ceRNA network consisting of 23 lncRNAs, 117mRNAs and 22miRNAs in lung cancer. Cao et al. also proposed a 6-lncRNA prognostic model. *Li et al. (2018)* constructed three ceRNA networks and a 32-ceRNA prognostic signature. However, they did not further explore the correlation between immune microenvironment and ceRNAs comprehensively. In the present study, we not only presented nomogram prediction models based on ceRNAs and immune cells, but also further investigated the relationships between crucial ceRNAs and immune microenvironment. GSE72094 dataset proved the feasibility of predicted equations.

Cigarette smoking is an established risk factor of lung cancer, which accounts for the leading cause of death worldwide (*Ettinger et al., 2019*). In recent years, the prevalence of LUAD has increased and smoking patients with LUAD constitute a certain proportion (*Zhang et al., 2015*).

Besides, we stratified TCGA patients with smoking history into three groups (smoking years >20, smoking years <=20 and non-smoking) and calculated risk scores in each sample via both predicted formulas, respectively. Not unexpectedly, both predicted formulas confirmed that the more extended smoking group (smoking years >20) have a higher risk than the shorter smoking group (smoking years <=20). Thus, it could be concluded that smoking was an adverse factor for prognosis in both models. Risk scores in the >20 years-smoking group were increased when compared with non-smoking groups in both models. However, there were no significant differences between ≤ the 20 years-smoking group and the non-smoking group. It was related to that the patients with less duration of smoking were more frequently female. E2F7, a member of EF transcription factors, was known as a transcriptional repressor that influences the prognosis in a lot of tumors, such as breast cancer, endometrial carcinoma, gallbladder cancer, and colon cancer. Previous studies have proved that miR-30a-5p prevents tumor migration and metastasis by targeting E2F7 in gallbladder cancer (*Ye et al., 2018*). *Liu et al. (2018)* have revealed that E2F7 was reported significantly overexpressed in ER-positive breast cancer than healthy breast tissues

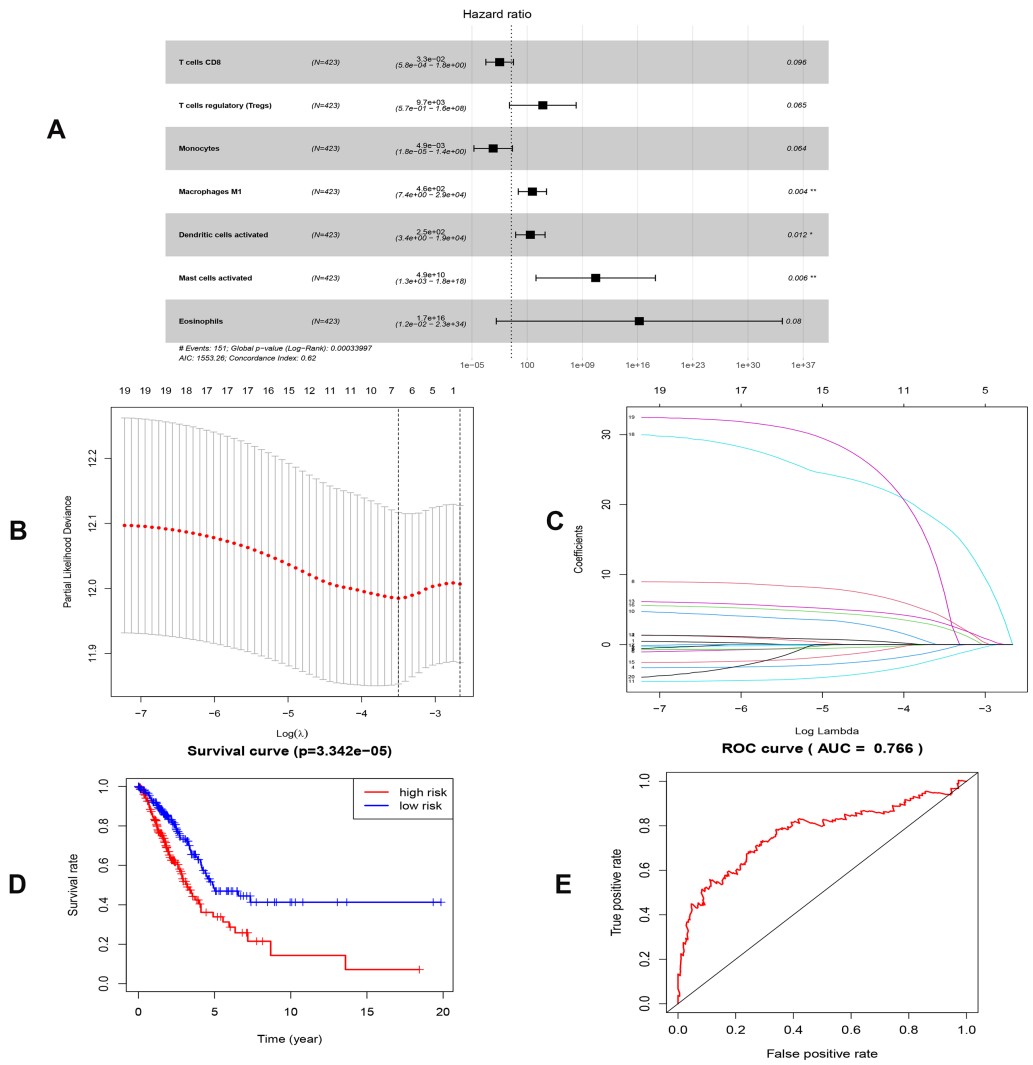

**Figure 8** **Construction of the prognostic model based on key immune cells in LUAD.** (A), multivariate Cox regression. (B), (C) lasso regression. (D) the Kaplan–Meier survival curve. (E) the ROC curve. ROC, the Receiver Operating Characteristic. LUAD, lung adenocarcinoma.

and then led to tamoxifen resistance in breast cancer cells. Additionally, overexpressed E2F7 could be suggested as a significant biomarker to identify the high-risk and low-risk groups of patients with lung cancer, which is also in consistence with our report (*Sun et al., 2018*).

SMOC1 is a cancer-associated protein, identified as an extracellular glycoprotein of the SPARC-related modular calcium-binding protein family. SMOC1 was overexpressed in brain cancer, including oligodendrogliomas, astrocytomas, and glioblastomas (*Brellier et al., 2011*; *Fackler et al., 2011*). However, till now, there is no enough evidence for the impact of SMOC1 in LUAD reported in the literature. Moreover, the current study may provide a study direction for researches of SMOC1 in the LUAD. Additionally, CCTA6 is reported to encode a molecular chaperone and play critical roles in damaged proteins
**Table 4    Multivariate Cox proportional hazards regression model based on immune cells in LUAD.**

| Immune cells | Coef | HR | 95% CI | *P* value |
|---|---|---|---|---|
| T cells CD8 | −3.426 | 0.033 | (5.767e−04-1.835) | 0.096 |
| T cells regulatory (Tregs) | 9.175 | 9653.370 | (0.571-1.631e+08) | 0.065 |
| Monocytes | −5.318 | 0.005 | (1.778e−05-1.351) | 0.064 |
| Macrophages M1 | 6.135 | 461.532 | (7.447-2.860e+04) | 0.004 |
| Dendritic cells activated | 5.539 | 254.456 | (3.446-1.879e+04) | 0.012 |
| Mast cells activated | 24.611 | 4.881e+10 | (1.320e+03-1.805e+18) | 0.006 |
| Eosinophils | 37.352 | 1.66589E+16 | (0.012-2.282e+24) | 0.080 |

**Notes.**
LUAD, lung adenocarcinoma; Coef, correlation coefficients; HR, hazrd ratio.

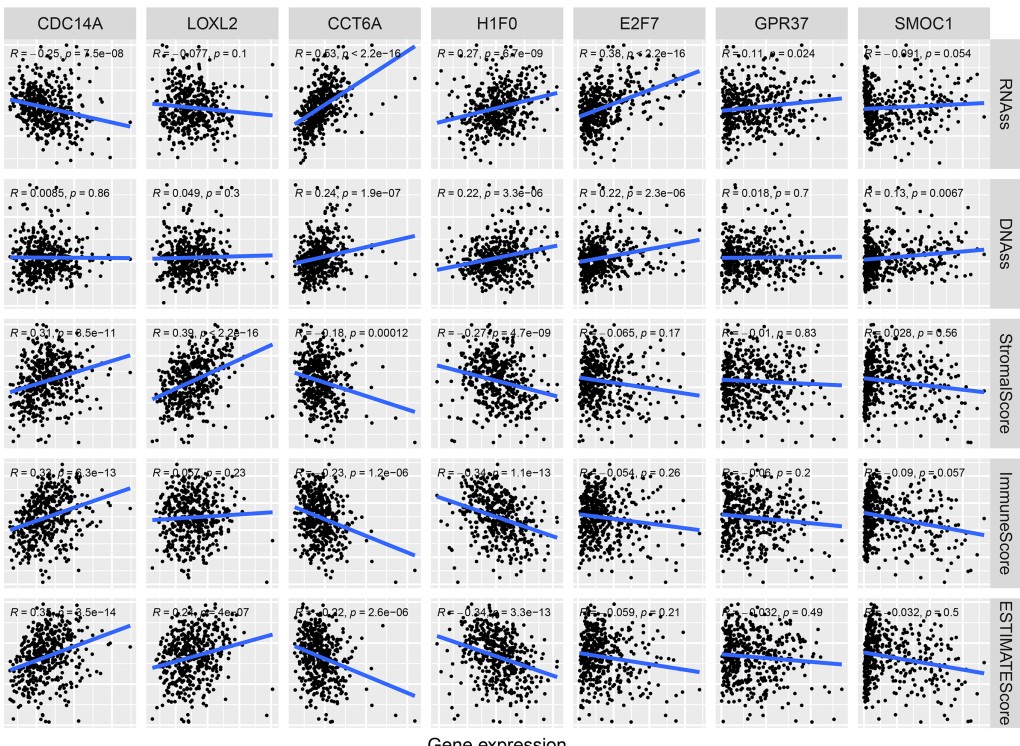

**Figure 9    Correlations between crucial ceRNAs expression and stem cell index, crucial ceRNAs expression and the stromal/immune/ESTIMATE score in LUAD.** RNAss and DNAss represent RNA-based stemness score, DNA-based stemness score, respectively. The sum of stromal score and immune score is ESTIMATE Score.

repair, cytoskeletal organization, and the cell cycle (*Van Hove et al., 2015*; *Yam et al., 2008*). *Ying et al. (2017)* found that CCTA6 served as a SMAD2-binding protein, which could inhibit the function of SMAD2, and promote metastasis.

CDC14A, which is of considerable significance to regulate the actin, was reported downregulated in many tumors and then reduced CDC14A was associated with the poor

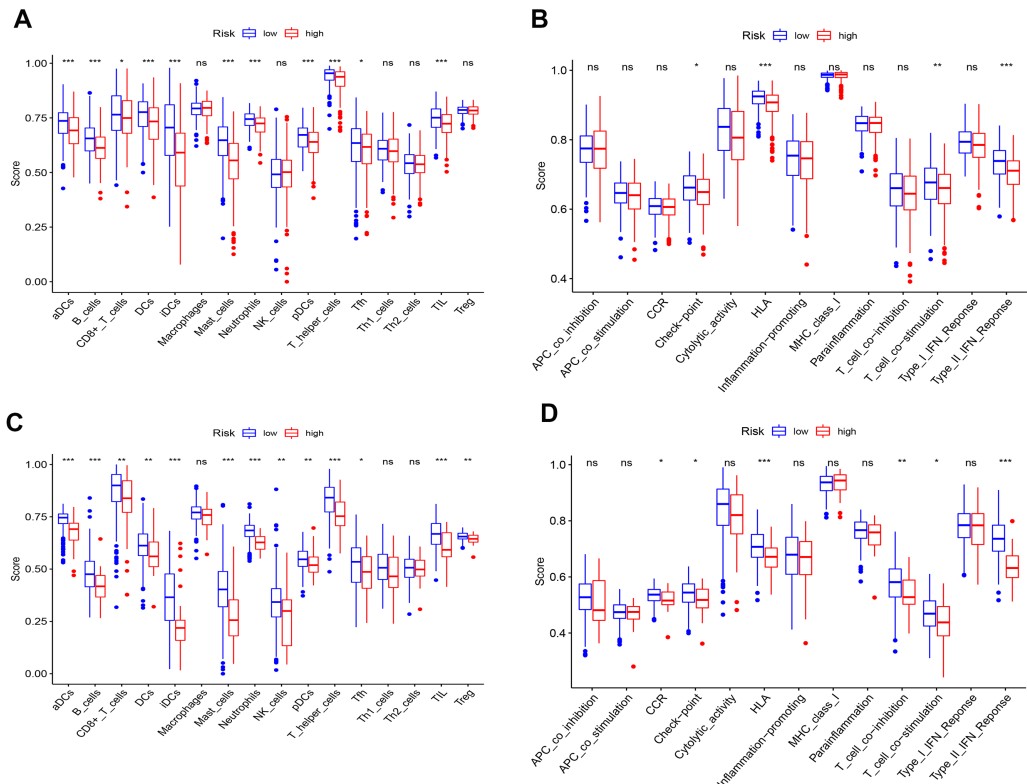

**Figure 10** **Differences of ssGSEA scores in high- and low- risk groups in TCGA (A–B) cohort and GEO (C–D) cohort.** (A & C) refer to the difference in immune cells. (B & D) refer to the difference in immune-associated functions * $p < 0.05$, ** $p < 0.01$, and *** $p < 0.001$.

clinical outcomes (*Chen et al., 2016*). Our results indicated that CDC14A was a protective biomarker in LUAD (HR=0.75, $P = 0.0066$), which corresponded well with previous studies. Similarly, based on the correlation with stemness scores and immune socres, we could also draw inferences that CDC14A suppressed tumor progression.

As one of H1 histone genes, H1F0 (known as H1.0) is heterogeneous in many tumors (*Di Liegro, Schiera & Liegro, 2018*). Knockdown of H1.0 influenced the differentiation of embryonic stem cells (*Terme et al., 2011*). Recent studies revealed that H1F0 were prognostic indicators in many cancers, such as breast cancer, liver cancer, and kidney cancer (*Torres et al., 2016*). In the current study, the relationships of H1F0 expression with stemness scores and Estamine scores suggested that H1F0 might promote LUAD.

T cells regulatory (Tregs) account for 5%–10% of peripheral CD4+ T cells in humans. Increasing evidence demonstrated that the immunosuppression mediated by Tregs is one of the essential mechanisms of immune evasion in tumors (*Dunn, Old & Schreiber, 2004*; *Shevach, 2002*; *Zou, 2005*). Mast cells are increased in several tumors, and their accumulation was associated with a low survival rate in many cancers, such as pancreatic adenocarcinoma (*Strouch et al., 2010*) and melanoma (*Ribatti et al., 2003*). Macrophages M1 was historically recognized as the proinflammatory subgroup,

expressing a series of chemokines and consequently play antitumor roles (*Hao et al., 2012*). Controversially, our results suggested that M1 was highly expressed in LUAD (Fig. 8). Actually, heterogeneity of environment signals could influence TAM development, and comprehensive nomenclatures of TAM have consequently been suggested (*Ostuni et al., 2015*).

Additionally, our study found that there were obvious differences in immune status between high- and low- risk groups in both TCGA and GEO cohorts. This indicated that decreased risk scores of ceRNAs were related to the enhancement of antitumor immunity. Furthermore, we could say that diminished immune function is responsible for poor prognosis in LUAD patients. However, there are inevitably some disadvantages in our study that should be taken into consideration. Firstly, considering the limit data from the TCGA and GEO, it could result in large analysis deviation and be verified by other large cohorts. Secondly, our follow-up experiment will explore the molecular mechanisms of prognostic signatures in the present study.

## CONCLUSION

To conclude, the present study firstly provided a combined analysis of ceRNA and immune cells and then established nomograms to predict the prognosis in LUAD reliably. Based on the present study result, we will carry out the biological experiments to validate key ceRNAs and mechanisms of immune cells in the future.

## ACKNOWLEDGEMENTS

The authors would like to thank the TCGA, GEO, TIMER, GEPIA, HPA, UCLAN and TISIDB databases.

### Funding

The authors received no funding for this work.

### Competing Interests

The authors declare there are no competing interests.

### Author Contributions

- Miaomiao Zhang conceived and designed the experiments, performed the experiments, analyzed the data, prepared figures and/or tables, authored or reviewed drafts of the paper, and approved the final draft.
- Peiyan Zheng performed the experiments, analyzed the data, prepared figures and/or tables, authored or reviewed drafts of the paper, and approved the final draft.
- Yuan Wang conceived and designed the experiments, analyzed the data, authored or reviewed drafts of the paper, and approved the final draft.
- Baoqing Sun conceived and designed the experiments, authored or reviewed drafts of the paper, and approved the final draft.
## Data Availability

All raw data are available at TCGA and GEO. TCGA IDs and sample GEO accession numbers are available in the Supplemental Files.

## Supplemental Information

Supplemental information for this article can be found online at http://dx.doi.org/10.7717/peerj.11029#supplemental-information.

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
