# Peer review of "Two predicted models based on ceRNAs and immune cells in lung adenocarcinoma"

_PeerJ, doi:10.7717/peerj.11029_

## Round 0.1 · original submission · Major Revisions

The manuscript has been reviewed by two referees and both have raised significant concerns. The revision should completely address all of these concerns.

·

Basic reporting

The contribution of this paper, in terms of novelty and performance improvement, is not enough.

Experimental design

No comment.

Validity of the findings

The introduction and discussion of variety of the algorithm principles used in this study and the responding results were insufficient.

Additional comments

This study constructed two cox regression models based on crucial ceRNAs and immune cells to predict prognosis in LUAD. This is a meaningful study, but there are several issues that the authors should address.
1. Two cox regression models based on crucial ceRNAs and immune cells respectively were constructed in this study, however, what the correlation of these two regression models were stated insufficient. In short, the contribution of this paper, in terms of novelty and performance improvement, is not enough.
2. In this study, ESTIMATE and CIBERSORT were mentioned to evaluate or estimate the proportion of stromal and immune cells, what’s the different between these this two methods, and why these two methods are both chosen should be clarified.
3. In addition, in the ‘Multiple databases validation’ part, UCLAN, GEPIA and TIMER (line 181) are also used on different databases; and in line 194, ssGSEA also as an algorithm for calculating the abundance of immune cells was used, I doubted why so many different immune cell infiltration related abundance-calculating methods were used on different datasets? If the authors want to compare the effectiveness of different algorithms, the results of different algorithms on the same dataset should be given, or the same algorithm is compared on different datasets.
A wide range of different datasets were used in this manuscript, at the same time, a variety of methods were used to calculate the abundance of immune cells such as CIBERSORT, ssGSEA, UCLAN, GEPIA and TIMER, how and why these different abundance calculating methods were selected was not stated. A sufficient discussion of the advantages and disadvantages of each method and why they are chosen should be provided.
4. For Figure 3 and Figure 7, the content in the figures should be described in detail, such as the meaning of the shapes, colors and size of the node, and so on.
5. This article uses many algorithms like CIBERSORT to calculate the abundance of immune cells, but the introduction and discussion of the algorithm principle and some necessary details of parameter setting for all the applied methods were missing.
6. A variety of datasets were used in the manuscript like mRNA, lncRNA, miRNA, for each dataset, however, the details including the number of samples and RNAs, information of patients and so on are missing.

Reviewer 2 ·

Basic reporting

Grammatical error in line 28, 169. spell check for DESeq in line 69
Figure legends need to be more elaborate

Experimental design

na

Validity of the findings

The authors have efficiently used the published dataset and tried to eludeidate a bunch of biological questions related to LUAD.

In performing the differential expression between normal and tumor
A. Did the authors compared their results to the results from 3 previous studies ; https://doi.org/10.1186/s12935-020-01295-8, https://doi.org/10.3892/ol.2018.9336, https://doi.org/10.3892/ijo.2016.3716, https://doi.org/10.1111/jcmm.15778
B. Looking at the heat maps of the DEG in fig1c - the 2 groups are not segregateing very well based on the clustering. If the genes were significantly different, I would expect clustering to separate them out.
C. A fold change of >|1| seems a bit lenient especially considering the fact effect sizes in tumor vs normal
D. For the dataset used in the first set- were any other co variants taken into account? Especially doing the cell type estimation

2. It is not clear what motivated the authors to look into the smoking status in nomograph predictions?
3. Is it possible for the authors to show some kind of clustering for the high vs. low risk groups transcriptomic profiles overlayed with any other important covariate?

4. More information is needed in the figure legends to make it more interpretable.

5. Also for the validation dataset, which seems to be a set of samples with different mutations, how was the dates used or conditioned for these mutations.

6. Now that there are tremendous amount of single cell transcriptomic data available, can the authors validate/look up some findings in any public datasets if available?

7. In fig 5E- except for a subset of samples- the distribution of the >20 and <20 smoking status look very similar. Did the authors look into what that subset is which is driving the most differences?

Additional comments

The authors in this manuscript used published dataset to identify differentially expressed mRNA,miRNA,lncRNA between normal tissue and tumor samples which were then used to construct a regulatory network of interactions. They further predicted patients prognostic values based on these findings. They also used CIBERSORT to estimate the immune cell types differences in normal vs. tumor and eventually integration into the multivariate model. They found the hub ceRNA to be associated with stemless and the microenvironment of which the key members were validated in another independent published dataset.

---

## Round 0.2 · accepted · Accept

The resubmission has been examined by one of the two reviewers who had looked at the previous version of the manuscript.

Reviewer 2 ·

Basic reporting

NA

Experimental design

NA

Validity of the findings

NA

Additional comments

The authors have addressed the concerns.